# Disrupted Gray Matter Networks Associated with Cognitive Dysfunction in Cerebral Small Vessel Disease

**DOI:** 10.3390/brainsci13101359

**Published:** 2023-09-22

**Authors:** Yian Gao, Shengpei Wang, Haotian Xin, Mengmeng Feng, Qihao Zhang, Chaofan Sui, Lingfei Guo, Changhu Liang, Hongwei Wen

**Affiliations:** 1Key Laboratory of Endocrine Glucose & Lipids Metabolism and Brain Aging, Ministry of Education, Department of Radiology, Shandong Provincial Hospital Affiliated to Shandong First Medical University, Jinan 250021, China; gaoyi_an@163.com (Y.G.); 15163693053@163.com (C.S.); 2Research Center for Brain-Inspired Intelligence, Institute of Automation, Chinese Academy of Sciences, Beijing 100040, China; wangshengpei2014@ia.ac.cn; 3University of Chinese Academy of Sciences, Beijing 101408, China; 4Department of Radiology and Nuclear Medicine, Xuanwu Hospital, Capital Medical University, No. 45 Chang-Chun St., Xicheng District, Beijing 100054, China; xiaotianyiyiyi@126.com (H.X.); fmm1462@163.com (M.F.); 5Department of Radiology, Weill Cornell Medical College, New York. 407 East 61st Street, New York, NY 10044, USA; qz288@cornell.edu; 6Department of Radiology, Shandong Provincial Hospital, Cheeloo College of Medicine, Shandong University, Jing-Wu Road No. 324, Jinan 250021, China; 7Key Laboratory of Cognition and Personality (Ministry of Education), Faculty of Psychology, Southwest University, Chongqing 400715, China

**Keywords:** cerebral small vessel disease, cerebral microbleeds, voxel-based morphometry, gray matter networks, graph theory

## Abstract

This study aims to investigate the disrupted topological organization of gray matter (GM) structural networks in cerebral small vessel disease (CSVD) patients with cerebral microbleeds (CMBs). Subject-wise structural networks were constructed from GM volumetric features of 49 CSVD patients with CMBs (CSVD-c), 121 CSVD patients without CMBs (CSVD-n), and 74 healthy controls. The study used graph theory to analyze the global and regional properties of the network and their correlation with cognitive performance. We found that both the control and CSVD groups exhibited efficient small-world organization in GM networks. However, compared to controls, CSVD-c and CSVD-n patients exhibited increased global and local efficiency (E_glob_/E_loc_) and decreased shortest path lengths (L_p_), indicating increased global integration and local specialization in structural networks. Although there was no significant global topology change, partially reorganized hub distributions were found between CSVD-c and CSVD-n patients. Importantly, regional topology in nonhub regions was significantly altered between CSVD-c and CSVD-n patients, including the bilateral anterior cingulate gyrus, left superior parietal gyrus, dorsolateral superior frontal gyrus, and right MTG, which are involved in the default mode network (DMN) and sensorimotor functional modules. Intriguingly, the global metrics (E_glob_, E_loc,_ and L_p_) were significantly correlated with MoCA, AVLT, and SCWT scores in the control group but not in the CSVD-c and CSVD-n groups. In contrast, the global metrics were significantly correlated with the SDMT score in the CSVD-s and CSVD-n groups but not in the control group. Patients with CSVD show a disrupted balance between local specialization and global integration in their GM structural networks. The altered regional topology between CSVD-c and CSVD-n patients may be due to different etiological contributions, which may offer a novel understanding of the neurobiological processes involved in CSVD with CMBs.

## 1. Introduction

Cerebral small vessel disease (CSVD) is a prevalent cerebrovascular disease that frequently occurs in conjunction with neurodegenerative disorders, and its structural imaging indicators include vascularized white matter hyperintensities (WMHs), perivascular spaces (PVS), and cerebral microbleeds (CMBs), etc. [1]. It progresses slowly and insidiously, resulting in the loss of numerous cognitive skills. The clinical symptoms of CSVD vary depending on the origin of the disease and the affected brain region, with cognitive impairment (CI), emotional or behavioral abnormalities, and movement disorders identified as possible manifestations in patients [2]. In addition, CSVD plays a crucial role in Alzheimer’s disease (AD). The accumulation of amyloid-beta (Aβ) in the brain leads to CI and affects white matter (WM) tracts, eventually resulting in the manifestation of clinical symptoms [3]. CMBs are perivascular deposits of blood breakdown products contained in macrophages [4], and they are linked to a number of risk factors, including advanced age, hypertension, and low cholesterol [5]. According to previous studies, they are still related to CI in symptomatic persons with CSVD after controlling for other imaging indicators [6].

Gray matter (GM) is mostly composed of neuron cell bodies and is an essential aspect of the central nervous system, which is closely related to cognitive function [7]. Gray matter volume (GMV) has been commonly applied in many studies as an essential measurement in recent years, and voxel-based morphometry (VBM) is an excellent tool that accurately evaluates GM alterations in brains [8,9]. Previous investigations using VBM methods documented that CSVD patients with mild cognitive impairment (MCI) and motor impairment have decreased GMV in multiple brain regions [10,11]; the atrophy of whole brain GMV is closely related to the damage of WM [12,13,14,15], and local WMH affected GMV in remote areas [16]. These studies have shown that GM in cognition-related brain areas exhibit morphological abnormalities in CSVD patients, but these modifications have not been able to explain the interactions and connections between the GM structures. Large-scale networks do include several different brain areas, each of which may have a unique pattern of interaction. The cognitive and attention systems, as well as symmetrically interhemispheric areas, have all been implicated in coordinated alterations in brain morphology across regions of functionally or physically related systems [17].

These days, research on neuropathological mechanisms frequently uses brain structure network analysis based on morphological metrics such as GMV [18,19,20,21]. GM networks were constructed using structural MRI by calculating GMV via VBM analysis and defining network edges based on the similarity of morphological distributions between brain regions [22]. The precise coordination of cortical morphology in the brain is reflected by morphometric correlations in the structural network, which are believed to be very similar to the direct anatomical link data acquired through tract tracing [23,24,25]. Common similarity metrics include the Kullback–Leibler divergence-based similarity measure [19] and structural covariance matrix measure [18]. Notably, the histogram-weighted metric is a statistical tool used to assess the similarity of morphology distributions across distinct regions of the brain, and it could provide valuable insights into the prediction of disease states. Its application in academic research has greatly improved our understanding of the brain morphology of neurological diseases [26]. The topological structure of the entire brain may be investigated once the GM network is built using statistical correlations of the morphological descriptors in conjunction with the brain network analysis approach, yielding thorough network-level data [18].

Graph theory analysis has grown in popularity recently in the disciplines of neuroimaging and brain network research, offering a sophisticated tool to examine the topological architecture of brain networks. Many important topological characteristics, including small-worldness and strongly linked hubs, are present in brain structural networks [27]. The use of graph theory analysis to investigate topological abnormalities in whole-brain structural networks of CSVD patients is significant and may provide valuable insights into underlying processes. According to a prior study, CSVD impaired the effectiveness of brain structural networks, which resulted in cognitive deterioration [28]. Individuals with AD had reduced network integrity in the posterior cingulate cortex and subcortical hub, which were connected to CSVD burden [29]. Novel evidence for compensatory mechanisms of the GM network was identified in AD patients, and the potential of applying structural network indicators to monitor disease progression was highlighted [30]. Although these previous studies on CSVD have exhibited a disruption of structural network efficiency, they were not independent of imaging markers of CSVD [28]. The alterations in brain network topological organization associated with the presence of CMBs in CSVD patients are still poorly understood.

To analyze the topological properties of both CSVD patients and healthy subjects, we used graph theory analysis to build brain structure networks using GM volumetric parameters. We predicted that the cognitive, attention and executive functional regions would undergo global and local topological remodeling in CSVD patients. Our understanding of the neurobiological mechanisms causing CSVD may be improved via this investigation, which, to the best of our knowledge, is the first effort to define the GM structural network of CSVD patients with various levels of subclinical damage (with or without CMBs). This study consists of five parts, namely the Introduction, Experimental Procedures, Results, Discussion and Conclusion.

## 2. Experimental Procedures

Section 2.1 will cover subject information. Section 2.2, Section 2.3, Section 2.4, Section 2.5 and Section 2.6 will cover imaging acquisition, and preprocessing for voxel-based morphometry, brain network construction, network topology analysis, and between-group statistical comparison and correlation analysis, respectively.

### 2.1. Subjects

The current cross-sectional study was approved by the Institutional Review Board at Shandong Provincial Hospital, affiliated with Shandong First Medical University. From December 2018 to December 2021, 49 CSVD patients with CMBs, 121 CSVD patients without CMBs, and 74 healthy subjects in total were recruited for this investigation. The study recruited healthy elderly volunteers aged 45 to 80 years from the local community. All participants had more than 7 years of education and underwent a comprehensive assessment of their cognitive functions. Current MRI consensus standards include the following criteria for CSVD: recent minor subcortical infarct diagnosis, lacunes of presumed vascular origin, WMH of presumed vascular origin, increased PVS, CMBs, and brain atrophy [1]. CMBs present as small hypointense lesions (2–5 mm) in T2*-weighted gradient-recalled echo or susceptibility-weighted sequences [31]. The severity of CSVD was administered using the Fazekas scale (0–3) for periventricular hyperintensity and deep white matter hyperintensity lesions [32] and via a combined simple CSVD score (the 0–3-point scale that was calculated based on the severity of CMBs, lacunes, and WMH) recommended recently for predicting cognitive decline [33].

The Montreal Cognitive Assessment (MoCA), a screening tool designed by Nasreddine et al. to detect MCI, was administered to all participants. The MoCA takes approximately 10 to 15 min to administer. The highest score is 30 points, and higher scores imply better cognition [34]. In addition, the participants’ language learning and memory abilities were assessed using the Rey Auditory Verbal Learning Test (AVLT) [35]. The Trail Making Test (TMT) measures graphomotor speed, visual scanning, and executive function [36], whereas the Stroop Color and Word Test (SCWT) measures conflict monitoring, working memory, and visual search speed [37]. In the cognitive assessment of information processing speed, which is essential for many cognitive operations, Symbolic Digital Modalities Testing (SDMT) is commonly applied [38]. The examiner had received professional training and certification but did not understand the subject groups.

The exclusion criteria and clinical parameters were as follows. Patients with the following conditions were excluded: (1) coronary atherosclerosis, heart disease; (2) atrial fibrillation, ventricular aneurysm, rheumatoid arthritis, vasculitis, drug addiction, and other probable causes of stroke; (3) carotid artery stenosis and neurological disorders such as AD, Parkinson’s disease, and epilepsy; (4) a well-known dangerous medical condition, such as cancer; (5) bilateral renal artery stenosis or chronic kidney disease in stages 4–5; and (6) MRI-related contraindications.

### 2.2. Image Acquisition

All participants underwent imaging using a 3.0 T MR scanner (MAGNETOM Skyra, Siemens Healthcare, Erlangen, Germany) equipped with a 32-channel head coil for receiving signals. We used the MPRAGE sequence to acquire 3D T1W images with a TR/TE of 7.3/2.4 ms, TI of 900 ms, FOV of 240 × 240 mm^2^, matrix size of 256 × 256, 192 slices, and slice thickness of 0.9 mm. No gap was used, and the flip angle was 9°. Susceptibility-weighted imaging (SWI) was conducted using a 3D T2*-weighted gradient echo sequence with a 1.5 mm slice thickness, 27/20 ms TR/TE, 220 × 220 mm^2^ FOV, and 256 × 256 matrix size. Before the scan, all subjects maintained their regular breathing and heart rates, as instructed, while remaining awake and in a state of quiet relaxation until the completion of the scan. T2W fluid-attenuated inversion recovery (FLAIR) sequences, T2-weighted (T2W) turbo spin echo sequences, SWI, and diffusion-weighted images were also acquired to detect brain abnormalities.

### 2.3. Preprocessing for Voxel-Based Morphometry

After image acquisition, all T1W images were first checked for scanner artifacts and gross anatomical abnormalities. Then, whole-brain VBM was preprocessed using statistical parametric mapping (SPM8, http://www.fil.ion.ucl.ac.uk/spm, accessed on 22 October 2021) software. VBM analysis usually consists of the following steps: (a) segmentation; (b) normalization; (c) modulation (Jacobian modulation), in which normalized gray matter maps are scaled via macroscopic transformations to preserve local volumes; and (d) smoothing. The New Segment tool in SPM was used to segment the GM, WM, and cerebrospinal fluid (CSF) in native space from the T1W images after they had been reoriented along the anterior–posterior commissure (AC-PC) line with the AC set as the coordinate origin. The sum of the GM, WM, and CSF volumes was used to determine the total intracranial volume (TIV). After that, a study-specific reference space based on diffeomorphic anatomical registration through exponentiated lie algebra (DARTEL) was created using all of the segmented GM images from all of the subjects [39] and then normalized into Montreal Neurological Institute (MNI) space at a 1.5 mm cubic resolution. Finally, the normalized GM images were modulated to ensure that relative GMV was preserved following spatial normalization. The modulated normalized GM images containing voxel-wise GMV were used for the subsequent generation of GM morphometric networks.

### 2.4. Network Construction

In this study, we constructed subject-wise structural networks from GM volumetric features using the graynet toolbox (https://github.com/raamana/graynet, accessed on 16 September 2021). Network nodes were defined based on the automated anatomical labeling (AAL) atlas, which parcellates the brain into 90 cortical and subcortical regions; please refer to Appendix A for details. Network edges were defined as the statistical similarity of morphological distributions between different brain regions [19]. The statistical similarity of morphological distributions between various brain areas was used to build network edges. According to the AAL atlas, GMV values were quickly collected from every voxel within each ROI, and the voxel-wise GMV distribution in a specific nodal area was then transformed into a histogram. The network connection was calculated as the histogram distance between two regions [40], which was characterized via the histogram intersection [26] metric in the histogram weighted networks (hiwenet) library (https://github.com/raamana/hiwenet, accessed on 16 September 2021). The mathematical definition of the histogram intersection metric is shown in Equation (1):(1)$$\mathrm{Histogram}\;\mathrm{intersection}(i,\;j)=\frac{\sum_{k=1}^{N}min(h_{i}\left(k\right),h_{j}(k))}{\sum_{k=1}^{N}h_{i}\left(k\right)}$$
where each region is indexed by i or j. For region i, *h_i_* is the normalized histogram of the voxel-wise GMV distribution. N is the number of bins in the histogram, which is fixed at N = 25 bins.

The brain structural network was created for each participant using a symmetric 90 × 90 network matrix created using T1-weighted MRI to create a person-specific GM morphological network, as illustrated in Figure 1. The network for each individual was thresholded according to the following standards, with the same sparsity ranging from 10% to 60% at an interval of 2%: the lower and upper bounds of the range were set in our previous study [41], which showed that the brain network became increasingly random and less biological above this bound. The minimum sparsity was used as the lower bound to avoid network fragmentation [21].

### 2.5. Network Topological Analysis

Graph theory analysis was employed to evaluate the global and regional network characteristics of the sparse networks obtained at each threshold. Five small-world property indicators and two network efficiency metrics were part of the global measures. Nodal betweenness centrality (BC) served as the regional indicator. Appendix A shows the general descriptions of the network properties.

To ensure comprehensive analysis, each topological property was determined at various sparsity thresholds. To summarize the results without focusing on a single threshold selection, we calculated the area under the curve (AUC) for each property across the entire range of sparsity values [41,42]. The hub and disrupted regions were identified using the nodal BC’s AUC value. The graph theoretical network analysis toolbox (GRETNA, http://www.nitrc.org/projects/gretna/, accessed on 14 March 2021) was used to implement graph theory analysis [43].

### 2.6. Between-Group Statistical Comparison and Correlation Analysis

After using Levene’s test to check the homogeneity of variance between groups, one-way analysis of variance (ANOVA) and the least significant difference (LSD) pairwise multiple comparison tests were conducted to compare age, education, cognitive test scores and TIV between groups. Meanwhile, the sex ratio, history of smoking and alcohol consumption, hypertension, hyperlipidemia, the presence of lacune, WMH, PVS, and CMBs between groups were compared using the chi-square test. With age, sex, education, and TIV as factors, ANCOVA was used to analyze differences between the three groups for global and nodal network metrics. LSD pairwise multiple comparison tests were used for pairwise comparisons. We used SPSS Version 24.0 (SPSS Inc., Chicago, IL, USA) to further assess the Pearson correlation coefficients between network metrics and cognitive parameters for all groups once significant intergroup differences in any network topological metrics had been detected. For all analyses, the significance level was set at *p* < 0.05.

## 3. Results

In Section 3.1, the subjects’ clinical and cognitive parameters are primarily discussed. Section 3.2, Section 3.3 and Section 3.4 provide information on the topological parameters and node changes of the GM brain network in patients with CSVD. Additionally, Section 3.5 examines the relationship between changes in network topology and cognitive parameters.

### 3.1. Demographic and Clinical Characteristics of the Subjects

The variances between groups of the demographic and cognitive characteristics were all homogeneous and detailed descriptive statistics are summarized in Table 1. The CSVD-c and CSVD-n groups had significantly lower Montreal Cognitive Assessment (MoCA), Rey Auditory Verbal Learning Test (AVLT), and SDMT scores and significantly higher Stroop Color and Word Test (SCWT) and Trail Making Test (TMT) scores than the control group, except for the lack of a significant difference in TMT scores between the CSVD-n and control groups. In addition, the CSVD-c group had significantly lower AVLT scores and significantly higher SCWT scores than did the CSVD-n group. No significant differences were observed in age, sex, education, or TIV among the three groups.

### 3.2. Alterations in the Global Properties of GM Networks in Patients with CSVD

All groups displayed a high-efficiency small-world topology with the properties γ > 1, λ ≈ 1, and σ = γ/λ > 1 across the entire sparsity range. Figure 2 shows that both CSVD-c and CSVD-n groups had significantly higher global and local efficiency (Eglob/Eloc) and lower shortest path lengths (Lp) than did control groups across various sparsity thresholds. Based on the findings in Table 2, no notable distinction emerged between the CSVD-c and CSVD-n groups. Nonetheless, both groups displayed considerably elevated AUC values for Eglob and Eloc, accompanied by decreased AUC values for Lp, in comparison to those of the control group. This implies that there were uniform and strong alterations across various sparsity thresholds. Other global characteristics did not significantly differ across the groups.

### 3.3. Partially Reorganized Hub Distributions of GM Networks among Groups

For each group, nodes were identified as pivotal brain hubs if their nodal BC exceeded the mean network BC by at least one SD [41]. Partially reorganized hub distributions were observed among the three groups, with nine common regions mainly located in the bilateral middle frontal gyrus (MFG), precuneus (PCUN), superior temporal gyrus (STG), left middle occipital gyrus (MOG), inferior parietal lobe (IPL) and inferior temporal gyrus (ITG). Compared with the CSVD-n and control groups, the CSVD-c group had one additional hub region in the left superior parietal gyrus (SPG) and lacked the right dorsolateral superior frontal gyrus (SFGdor) and median cingulate gyrus (DCG) as hub regions. In Table 3 and Figure 3, it is shown that both groups with CSVD had no left middle temporal gyrus (MTG) as a hub region. However, the group with CSVD-n had an additional hub region, the left triangular inferior frontal gyrus (IFGtriang).

### 3.4. Alterations in the Regional Properties of GM Networks in Patients with CSVD

Twelve brain regions demonstrating significant changes in BC were identified through ANCOVA tests with a *p*-value of less than 0.05. Table 4 displays alterations within functional modules associated with DMN, sensorimotor processing, attention, and visual processing observed across the three groups [44]. Additionally, pairwise intergroup differences were found using LSD multiple comparison testing. The CSVD-c group showed significantly increased nodal BC in the bilateral anterior cingulate gyrus (ACG) as compared to the CSVD-n and control groups, right MTG and left SPG, and decreased nodal BC in the left SFGdor. In addition, Table 4 and Figure 4 reveal that the CSVD-n group exhibited significantly higher nodal BC in the left IPL and insula. However, they displayed lower nodal BC in the left fusiform gyrus, right opercular inferior frontal gyrus, and caudate nucleus relative to those of the control group.

### 3.5. Correlations between Network Topological Alterations and Cognitive Parameters

Figure 5 displays the results of Pearson’s correlation analysis, which revealed significant correlations (*p* < 0.05, FDR corrected) between global topological properties (Eglob, Eloc, and Lp) and SDMT scores in both CSVD-c and CSVD-n patients, but not in the control group. In contrast, Figure 6 illustrates that the global properties of the control group were significantly correlated with the MoCA, AVLT, SCWT, and TMT scores but not the SDMT scores.

## 4. Discussion

In this study, we obtained several important findings, as described below. (1) Although both the CSVD and control groups exhibited the small-world structure at varying sparsity levels, the CSVD groups displayed significantly increased E_glob_/E_loc_ and decreased L_p_ compared with those of the control group. (2) Although the three groups showed similar hub distributions, CSVD-c patients exhibited significantly altered nodal BC in the DMN and sensorimotor-associated functional regions compared with those of the CSVD-n patients and controls. (3) Significant correlations were observed between E_glob_, E_loc,_ and L_p_ and five cognitive domains and explained their clinical significance, which may advance our understanding of the neurobiological mechanisms underlying CSVD. To our knowledge, this study represents the first attempt to use graph theory methodology to analyze the GM structural network of CSVD patients with and without CMBs.

The presence of the small-world topology in structural brain networks of both CSVD patients and controls was discovered in this study, suggesting that brain structural networks are reliably constructed. Small-world networks combine the strong clustering of regular networks with the short path lengths of random networks [45] to reduce wiring costs and facilitate information flow [46]. E_glob_, E_loc_, and L_p_ are important metrics for describing the information transmission efficiency of a network. L_p_ and E_glob_ are used to determine the global transmission capacity of a network [27]. E_loc_ symbolizes its ability to guard against random attacks to some extent. Changes in these parameters indicated a disrupted topology of the structural network in CSVD patients. We discovered that compared with the controls, the CSVD-c and CSVD-n patients had increased E_glob_ and E_loc_, as well as decreased L_p_. A shorter L_p_ and higher E_glob_ indicate stronger overall information integration capacity in the brain network, while a higher E_loc_ reflects stronger local information processing capacity and network resilience against attacks [21,45]. These findings suggest that compared to the normal control group, both groups of CSVD patients underwent certain optimizations in their GM brain networks. According to previous studies, structural network efficiency is a form of brain reserve, and improvements in structural network efficiency may prevent clinical deterioration in the presence of brain pathology [47]. By further comparing the brain network parameters of the two groups of CSVD patients, it was observed that although there were no significant differences in E_glob_, E_loc_, and L_p_, the CSVD-n patients showed a more favorable trend toward the optimization of the brain network. This may be attributed to the fact that CMBs still have a detrimental effect on the GM structural network of the brain, thereby impacting the optimization process of the brain network. Research has indicated that the pathogenesis of CMBs primarily involves impaired endothelial function and a disruption of the blood–brain barrier, both of which lead to structural and functional damage in endothelial cells. These changes promote the occurrence of vascular inflammation, ultimately resulting in dysfunction in self-regulatory mechanisms [48]. Therefore, the GM structural network of CSVD patients was adaptively reorganized and optimized to maintain normal brain function. By improving E_glob_ and E_loc_, the topology of the brain structure network was more optimized than that of the normal group.

Brain network hubs refer to brain parts that are more densely connected than others are. Hubs with high centrality help with long-distance communication [49], which is essential for effective communication. We discovered that global hub distributions in the three groups were relatively similar, suggesting that these critical brain parts structure are preserved throughout development and that small-world networks may tolerate developmental abnormalities or diseases [27]. The SFG is associated with cognitive and motor control activities; it is a key part of the dorsolateral prefrontal cortex and is strongly associated with depression [50]. The DCG is associated with a number of emotion-related functions [51], and the right DCG is engaged in monitoring conflicts to promote task-relevant actions [52]. The MTG is very important for semantic control and understanding visual information [53,54]. Furthermore, the SPG is a functionally diverse region; in addition to being involved in visuospatial and visuomotor integration [55], it plays roles in attention [56], written language [57], and working memory [58]. Another study showed that the left SPG helps distinguish AD from MCI [59]. The IFG has been implicated in gesture–speech integration [53], especially the IFGtriang, and the brain is more sensitive to the semantic relationship between hand movements and speech [60]. These findings provide substantial evidence for our study. Because of the existence of brain reserves and compensation mechanisms that can actively compensate for clinical deterioration in individuals with pathological conditions [47], we can reasonably speculate that in this study, the increased left SPG nodes in CSVD-c patients could compensate for the decreased right SFGdor, right DCG and left MTG. Similarly, the added left IFGtriang in patients with CSVD-n might also compensate for the function of the left MTG in semantic understanding.

In our research, we also discovered distributed regions with altered nodal efficiency in the CSVD-c group compared with those in the CSVD-n and control groups, and the involved regions were categorized into DMN and sensorimotor association functions. First, the disrupted regions in the DMN mainly involved the left SFGdor, bilateral ACG, and right MTG. The DMN is a network of brain regions that are more active during rest than during the performance of many attention-demanding tasks [61]. ACG is one of the noncore nodes in the DMN [62], and it plays roles in cognition and regulating emotion [63]. Activity in brain parts associated with attention control, such as the dorsal anterior cingulate gyrus, is reduced before the loss of attention [61]. Additionally, in another study, the increased nodal efficiency of the bilateral ACG was considered an index that reflects a resolution from overt hepatic encephalopathy (OHE) [64], and it may compensate for the loss of attention and execution function induced by OHE. In our study, patients in the CSVD-c group were categorized based on the location of CMBs into three subtypes: lobar (cortical and subcortical regions); deep (basal ganglia, thalamus, internal capsule, external capsule, corpus callosum, and deep and periventricular white matter); and infratentorial areas (brainstem and cerebellum) [65,66]. Hence, we divided them into three subgroups of CMBs: the lobar type (participants had CMBs located in lobar regions), the deep type (participants had CMBs located in deep and/or infratentorial regions), and the mixed type (participants had CMBs located in both lobar and D/I regions). The lobar subtype was the most prevalent and was unevenly distributed in the left and right cerebral hemispheres. Frontal lobe dysfunction has been reported to be more common in the lobar type than in the deep type [67], and the medial prefrontal cortex and the anterior cingulate all belong to the important node of the DMN [68,69]. As a result, we safely assume that the increased B_nodal_ in the ACG compensated for the loss of attention and executive function caused by CSVD-c. The MTG is proposed to play a role in language-related tasks such as lexical comprehension and semantic cognition, and it also underlies the role of the DMN in language function [70]. According to a previous study, patients with temporal tumors show a higher node efficiency in the left MTG, leading to altered memory function in patients [71]. This result may suggest that an increased nodal BC of the MTG may also affect the memory function of CSVD-c patients.

In addition, increased nodal BC was observed in the left SPG, which belongs to the sensorimotor functional modules and is critical for the execution of voluntary movements [72]. Data from a previous study showed that the SPG is functionally connected to the sensory, motor, and associative cortical regions [73], and AD patients showed decreased nodal efficiency in the left SPG, which is considered one of the potential causes of CI in AD patients [74]. According to the compensatory mechanism of the brain network [47,75], we inferred that the B_nodal_ of the left SPG was increased in CSVD-c patients to compensate for the normal operation of motor function. Previous studies have shown that a disruption of the blood–brain barrier leads to immune cell infiltration and inflammation, which is one of the important mechanisms of CSVD [76]. CMBs, one of the manifestations of CSVD, themselves lead to a sustained local inflammatory response, as manifested by the initial activation and continuous increases in activated microglia and macrophages [77]. All inflammatory markers were present at high levels in the CMBs [78]. In addition, CMBs may trigger toxic biological cascades with long-term effects on the brain parenchyma [79]. Therefore, the pathological changes in CSVD-c patients are more severe than those in CSVD-n patients, which may explain the difference in nodal efficiency between the CSVD-c and CSVD-n groups. In this research, the lobar subtype was the most prevalent and was unevenly distributed in the left and right cerebral hemispheres. Previous research has demonstrated that among the four clinical subtypes of Alzheimer’s disease, the logopenic variant of primary progressive aphasia (lvPPA) has the highest prevalence of CMBs (50%), primarily localized in the left hemisphere. Additionally, a significant reduction in cerebral blood flow volume is observed in the left hemisphere of patients with lvPPA. Left predominant hypoperfusion may arise as a consequence of anomalous microcirculation resulting from the presence of lobar CMBs and cerebrovascular injuries [80]. Increasing evidence suggests a significant association between the radiological manifestations of CSVD and the pathological features of dementia and AD [81]. Hence, it is reasonable to assume that in this study, the higher prevalence of large CMBs among patients with the lobar subtype and their unbalanced distribution resulted in a significant reduction in the B_nodal_ of CAU. L when compared to that of the HC group. This confirms that CMBs can disrupt the topological structure of the GM network in the brain. However, the sample size of the CSVD-c patients in this study was relatively small. In the future, we will further expand the sample size to further explore the specific mechanism of brain injury caused by CMBs. In addition, previous studies have found that the right caudate nucleus is larger than the left caudate nucleus in the healthy control group [82,83,84]. The caudate nucleus is closely associated with cognitive function, and a smaller volume of the caudate nucleus is linked to poorer cognitive function [85]. In this study, it was found that CSVD-n patients have decreased nodal efficiency in the CAU. R compared to normal participants. This could possibly be attributed to the fact that the right caudate nucleus in normal participants is naturally larger in volume. However, in CSVD patients, the disease disrupts the normal structure of the caudate nucleus, leading to a significant decrease in nodal efficiency in the right caudate nucleus.

In addition, we also performed a correlation analysis of E_glob_, E_loc_, and L_p_ with five cognitive domains. In accordance with compensation theory [47,75], we can speculate that the more severe the symptoms of CSVD patients, the more efficient the structural network required to maintain normal brain function. We could infer that under normal circumstances, E_glob_ and E_loc_ are negatively correlated with MoCA, AVLT, and SDMT scores and positively correlated with SCWT and TMT scores. L_p_, on the other hand, was positively correlated with MoCA, AVLT, and SDMT scores and negatively correlated with SCWT and TMT scores. Our study supports this conclusion. Increased Aβ promotes neuronal dysfunction and network alterations in learning and memory circuitry prior to the clinical onset of AD, leading to CI, according to previous research [86], while the occurrence of CMBs is also related to the accumulation of amyloid deposits [80]. Therefore, we can infer that the pathological cause of CMBs changes the correlation between the organizational efficiency of the brain network and cognitive function so that the brain network efficiency of CSVD patients is not significantly correlated with MoCA, AVLT, and SCWT scores. However, another study showed that vascular MCI patients had the highest incidence of impairment in the cognitive domain of the SDMT, which means that the speed at which the brain processes information was affected, and this impairment was associated with significantly reduced GMV in the frontal regions [87], providing a rationale for our reasoning. We found that the E_glob_ and E_loc_ of the disease group were significantly negatively correlated with SDMT. It can be considered that the increased network efficiency compensated for the function of information processing-related brain regions, thereby ensuring the normal operation of the brain. Our findings regarding the correlations between network topological properties and SDMT may provide neuroimaging evidence for cognitive impairment in CSVD patients.

Our team conducted a thorough investigation of the resting-state functional connectivity (FC) network of patients with CSVD patients. We compared our findings to a current study on the brain’s GM structural network of CSVD patients and discovered that the brain network construction models used in the two studies differed. This study had a larger sample size than did the previous one. Our research revealed that CSVD patients had a decreased global and local efficiency in their FC network and an increased shortest path length, while the opposite was observed in their GM structural network. We hypothesized that this was due to CSVD patients’ reduced brain functional efficiency caused by their condition. Consequently, their GM structural network underwent adaptive optimization and reorganization to maintain normal brain function. The topology of their brain GM structure network was more optimized than was that of normal subjects. Our study complements previous research and provides reliable results, expanding our understanding of the brain network of GM structure in CSVD patients.

Several limitations should be acknowledged in the present study. First, the small sample size of individuals with CSVD-c and the cross-sectional study design restricts the generalizability of the findings. It is crucial to increase the sample size of CSVD-c and conduct typing and subgrouping studies in future research. Additionally, longitudinal studies are warranted to directly assess the impact of CSVD progression on the dynamic changes in brain structural network topological attributes. Second, it is important to note that this study was conducted within a limited patient population at a research institute. To ensure the universal applicability and accuracy of our research results, it is imperative to include CSVD subjects from a wider range and comprehensively collect clinical and imaging data. Last, the VBM method employed in this study measures a combination of cortical gray matter characteristics, such as cortical surface area, cortical folding, and cortical thickness [88]. Hence, further morphological assessments specific to GM degeneration, such as measurements of cortical thickness, must be employed to investigate more accurate modifications in GM networks.

## 5. Conclusions

In conclusion, structural brain network analysis provides researchers with a more comprehensive perspective to explore the mechanism of CI in patients with do CSVD than traditional imaging markers and brain network properties have benefits over traditional imaging markers. With the gradual deepening of large-scale brain network analyses of CSVD patients, more studies have shown that the deterioration of the properties of structural and functional networks is strongly related to CI in CSVD patients, providing a basis for the early detection and diagnosis of CSVD. These findings offer a complete perspective on the reorganization of structural networks between regions that are linked to the pathophysiology of CSVD with CMBs.

## Figures and Tables

**Figure 1 brainsci-13-01359-f001:**
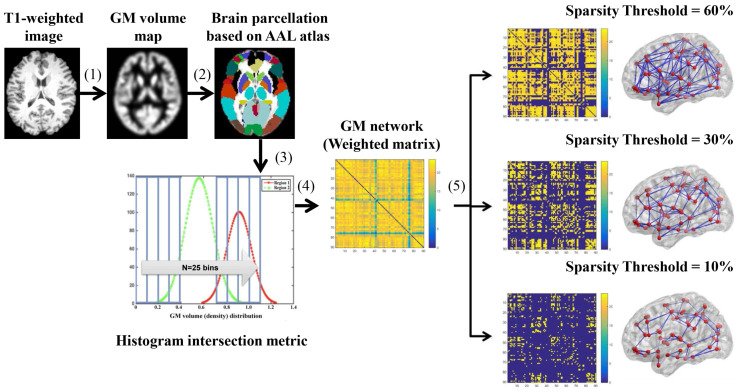
Flowchart for the construction of subject-wise GM morphological networks using T1-weighted MRI. (1) GM volume maps were created by segmenting, normalizing, modulating, and smoothing individual structural pictures using the VBM-DARTEL method. (2) GM volume map partitioned into 90 regions based on the AAL atlas. (3) Voxel-wise GMV for each region, retrieved and utilized to generate a histogram, and (4) the determined histogram distance (histogram intersection metric [26]) between each pair of areas, resulting in a 90 × 90 similarity matrix. (5) An interval of 2% and a sparsity range from 10% to 60% used to threshold each matrix. Sparse network visualization is shown in the lateral perspective.

**Figure 2 brainsci-13-01359-f002:**
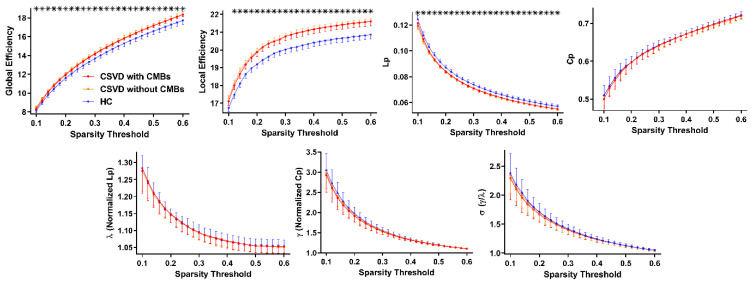
Group comparisons of global topological properties among the three groups. Data points indicated with a star indicate the global network metric demonstrating significant differences between the CSVD-c/CSVD-n group and the control group under a corresponding sparsity threshold (*p* < 0.05, ANCOVA with LSD post hoc test). The CSVD-c and CSVD-n groups showed no differences.

**Figure 3 brainsci-13-01359-f003:**
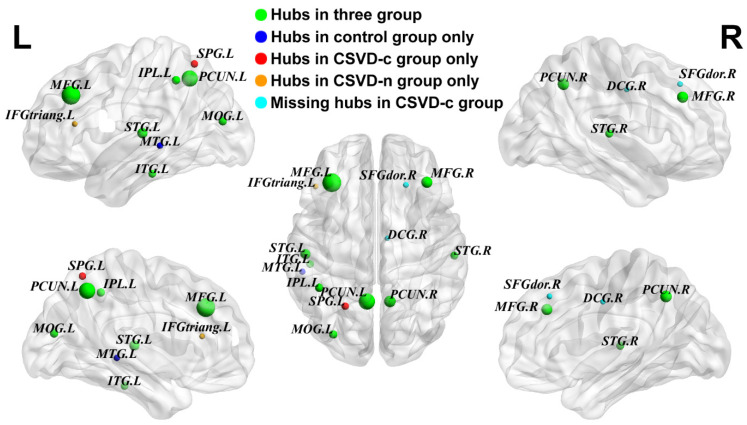
Hub region distributions of the GM networks for both groups. The hub nodes’ various node sizes are displayed alongside their nodal betweenness centrality scores. Software called BrainNet Viewer V1.6 (http://www.nitrc.org/projects/bnv/, accessed on 12 May 2021) was used to display the brain graphs. Please see Appendix A for a list of node acronyms.

**Figure 4 brainsci-13-01359-f004:**
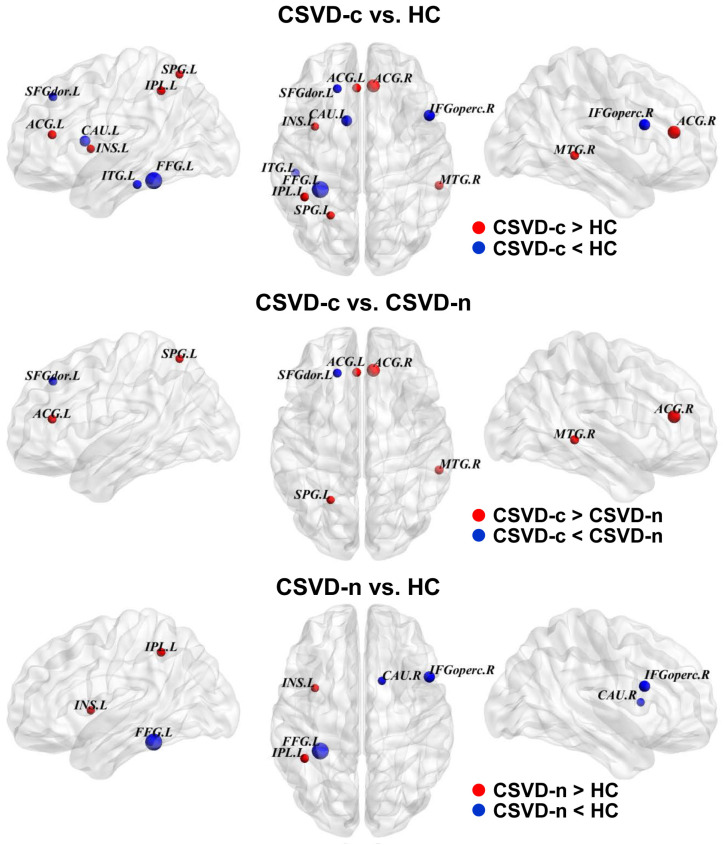
The differences in nodal betweenness centrality of the GM networks among the three groups. The scaled node sizes represent the F values in the ANOVA, and the disrupted nodes with substantially altered nodal betweenness centrality are represented in blue or red, respectively. Please see Appendix A for a list of node acronyms.

**Figure 5 brainsci-13-01359-f005:**
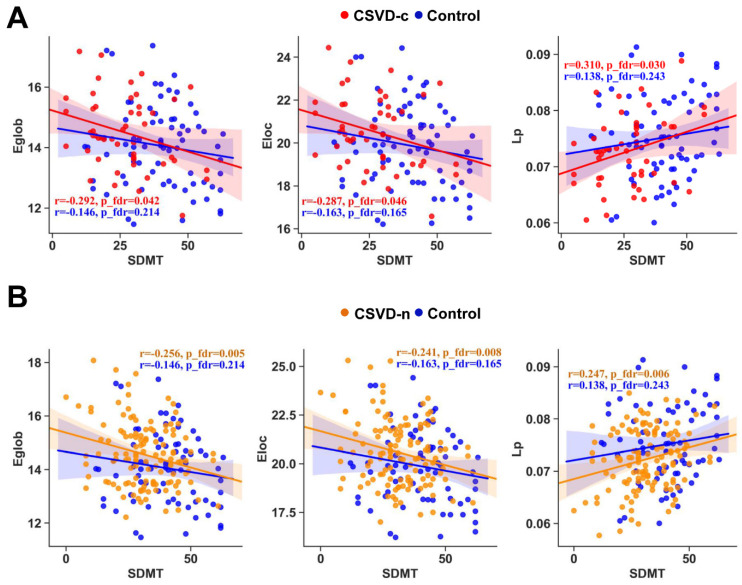
Scatter plots showing the significant Pearson’s correlations between network topological properties (Eglob, Eloc, and Lp) and SDMT scores for the (**A**) CSVD−c (red) and (**B**) CSVD−n (orange) groups. One subject is indicated with each dot. We give linear regression lines, r (partial correlation coefficient), *p* values (FDR adjusted), and 95% confidence intervals for the best−fit line (shading area). In contrast, the scatter plots of the control group (blue) are also shown in each subgraph.

**Figure 6 brainsci-13-01359-f006:**
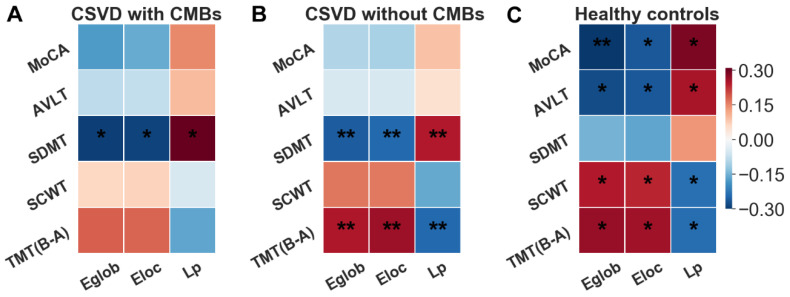
Pearson’s correlation coefficients between the network topological properties and cognitive parameters in both the CSVD and control groups. This heatmap displays the correlation coefficient between global/regional metrics and cognitive test scores for the CSVD-c (**A**), CSVD-n (**B**), and control (**C**) groups. *: *p* < 0.05, **: *p* < 0.01. Higher scores on the TMT (B-A) and SCWT were associated with worse symptoms. Conversely, higher scores on the MoCA, AVLT, and SDMT were associated with better symptoms. For both groups, global/local efficiency (E_glob_, E_loc_) was negatively correlated with MoCA, AVLT, and SDMT scores and positively correlated with SCWT and TMT (B-A) scores, while shortest path length (L_p_) was the reverse. For the abbreviations of nodes, please see Appendix A.

**Table 1 brainsci-13-01359-t001:** Demographic and cognitive characteristics of CSVD patients and controls.

Characteristic	CSVD-c (*n* = 49)	CSVD-n (*n* = 121)	HC (*n* = 74)	*p* Value (ANOVA/χ^2^)	*p* Value (Post Hoc)		
					CSVD-c vs. HC	CSVD-c vs. CSVD-n	CSVD-n vs. HC
Sex, female (%)	19 (38.8%)	59 (48.8%)	41 (55.4%)	0.196 ^χ2^	-	-	-
Age (y)	63.69 ± 8.37	63.08 ± 7.73	60.85 ± 9.05	0.096 ^a^	-	-	-
Education (y)	11.51 ± 2.94	11.77 ± 3.24	12.68 ± 3.59	0.094 ^a^	-	-	-
Smoke	19 (38.8%)	26 (21.5%)	19 (25.7%)	0.067 ^χ2^	-	-	-
Alcohol	24 (49.0%)	30 (24.8%)	20 (27.0%)	0.006 ^χ2^	0.002	0.013	-
Hypertension	26 (53.1%)	63 (52.1%)	33 (44.6%)	0.534 ^χ2^	-	-	-
Hyperlipidemia	25 (51.0%)	48 (39.7%)	28 (37.8%)	0.300 ^χ2^	-	-	-
Lacune	16 (32.7%)	23 (19.0%)	0	0.055 ^χ2^	-	-	-
WMH	47 (95.9%)	108 (89.3%)	0	0.165 ^χ2^	-	-	-
PVS	32 (65.3%)	48 (39.7%)	0	0.002 ^χ2^	-	-	-
CMBs	49 (100.0%)	0 (0.0%)	0	<0.001 ^χ2^	-	-	-
CMBs-lobar	23 (46.9%)	-	-	-	-	-	-
CMBs-deep	18 (36.7%)	-	-	-	-	-	-
CMBs-mixed	8 (16.3%)	-	-	-	-	-	-
MoCA	24.34 ± 3.05	25.23 ± 3.65	26.51 ± 3.58	0.003 ^a^	0.001	0.143	0.015
AVLT	55.81 ± 14.87	60.37 ± 11.31	64.51 ± 11.89	0.001 ^a^	<0.001	0.032	0.024
SDMT	27.43 ± 12.31	31.32 ± 11.94	40.01 ± 13.37	<0.001 ^a^	<0.001	0.071	<0.001
SCWT	169.15 ± 58.97	151.10 ± 45.11	133.32 ± 30.52	<0.001 ^a^	<0.001	0.019	0.008
TMT(B-A)	152.51 ± 97.74	130.64 ± 101.03	106.00 ± 80.72	0.030 ^a^	0.009	0.182	0.083
TIV	1.60 ± 0.15	1.57 ± 0.14	1.61 ± 0.16	0.143 ^a^	-	-	-

Notes: Data are presented as the mean ± SD. ^χ2^: chi-square test; ^a^: ANOVA test. For the abbreviations, please see Appendix A.

**Table 2 brainsci-13-01359-t002:** Group comparisons of AUC values of global network properties.

Global Property (AUC Value)	CSVD-c	CSVD-n	HC	*p* Value (ANCONA)	*p* Value (Post Hoc)		
					CSVD-c vs. HC	CSVD-c vs. CSVD-n	CSVD-n vs. HC
E_glob_	14.50 ± 1.20	15.56 ± 1.21	14.03 ± 1.36	0.014	0.041	0.792	0.005
E_loc_	20.54 ± 1.67	20.58 ± 1.72	19.84 ± 1.91	0.015	0.034	0.906	0.006
L_p_ (×e^−2^)	7.27 ± 0.62	7.23 ± 0.58	7.52 ± 0.72	0.006	0.031	0.690	0.002
C_p_ (×e^−1^)	6.45 ± 0.08	6.45 ± 0.08	6.46 ± 0.08	0.545	-	-	-
γ	1.59 ± 0.09	1.61 ± 0.10	1.61 ± 0.10	0.521	-	-	-
λ	1.11 ± 0.02	1.10 ± 0.02	1.11 ± 0.02	0.836	-	-	-
σ	1.42 ± 0.08	1.43 ± 0.09	1.44 ± 0.09	0.462	-	-	-

Note: Data are presented as the mean ± SD.

**Table 3 brainsci-13-01359-t003:** Hub regions of GM networks in both the CSVD and control groups.

CSVD with CMBs		CSVD without CMBs		HC	
Regions	B_nodal_	Regions	B_nodal_	Regions	B_nodal_
MFG.L	77.27 ± 37.42	MFG.L	75.24 ± 44.94	MFG.L	69.23 ± 37.91
MFG.R	53.95 ± 34.08	MFG.R	53.92 ± 39.83	MFG.R	50.63 ± 37.49
MOG.L	48.07 ± 32.85	MOG.L	47.13 ± 34.00	MOG.L	39.60 ± 26.70
IPL.L	51.30 ± 32.29	IPL.L	47.46 ± 39.94	IPL.L	36.51 ± 25.35
PCUN.L	66.59 ± 34.33	PCUN.L	68.01 ± 33.36	PCUN.L	66.99 ± 37.05
PCUN.R	56.46 ± 38.80	PCUN.R	52.27 ± 34.29	PCUN.R	53.21 ± 31.44
STG.L	48.73 ± 31.73	STG.L	47.90 ± 28.32	STG.L	53.63 ± 29.89
STG.R	44.87 ± 23.02	STG.R	46.12 ± 30.12	STG.R	39.84 ± 31.54
ITG.L	36.99 ± 24.88	ITG.L	43.25 ± 28.14	ITG.L	50.17 ± 29.77
SPG.L	42.54 ± 40.94	SFGdor.R	38.09 ± 29.21	SFGdor.R	37.34 ± 22.37
		IFGtriang.L	37.33 ± 31.29	DCG.R	37.56 ± 25.24
		DCG.R	35.46 ± 24.02	MTG.L	39.52 ± 24.86

Abbreviations: B_nodal_ represents the AUC value (mean ± SD) of nodal betweenness centrality across thresholds. For the abbreviations of nodes, please see Appendix A.

**Table 4 brainsci-13-01359-t004:** Brain regions showed significantly altered nodal betweenness centrality among the three groups for GM networks.

		B_nodal_			*p*-Value (ANCONA)	*p*-Value (Post Hoc)		
Module	Region	CSVD-c	CSVD-n	Control		CSVD-c vs. HC	CSVD-c vs. CSVD-n	CSVD-n vs. HC
DMN	SFGdor.L	17.99 ± 15.30	27.23 ± 25.47	28.00 ± 24.18	0.039	0.021	0.020	N.S.
DMN	ACG.L	21.1 ± 24.39	13.72 ± 13.54	14.45 ± 17.33	0.038	0.038	0.013	N.S.
DMN	ACG.R	20.3 ± 18.15	13.60 ± 14.12	12.36 ± 12.91	0.009	0.004	0.008	N.S.
DMN	MTG.R	26.24 ± 23.96	18.92 ± 18.12	17.74 ± 16.29	0.035	0.015	0.023	N.S.
attention	IFGoperc.R	16.64 ± 15.54	16.50 ± 15.18	23.62 ± 21.85	0.016	0.032	N.S.	0.006
attention	IPL.L	51.30 ± 32.63	47.46 ± 40.11	36.51 ± 25.52	0.038	0.022	N.S.	0.034
attention	ITG.L	36.99 ± 25.14	43.25 ± 28.26	50.17 ± 29.97	0.038	0.012	N.S.	N.S.
sensory/motor	INS.L	17.52 ± 16.61	16.35 ± 17.03	11.45 ± 11.18	0.048	0.033	N.S.	0.032
sensory/motor	SPG.L	42.54 ± 41.36	31.24 ± 32.10	27.53 ± 30.30	0.048	0.016	0.048	N.S.
subcortical	CAU.L	6.96 ± 8.75	11.18 ± 12.34	14.63 ± 20.13	0.018	0.005	N.S.	N.S.
subcortical	CAU.R	15.92 ± 17.28	15.02 ± 16.06	20.57 ± 17.75	0.047	N.S	N.S.	0.015
vision	FFG.L	7.40 ± 7.14	8.86 ± 10.69	14.55 ± 15.13	0.001	0.001	N.S.	0.001

Abbreviations: B_nodal_ represents the AUC values (mean ± SD) of the nodal betweenness centrality; N.S., not significant. The modular division of brain regions was based on a previous study [44]. For the abbreviations of nodes, please see Appendix A.

## Data Availability

The original contributions presented in the study are included in the article/Appendix A, further inquiries can be directed to the corresponding author.

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
