# Peer review of "Disrupted Gray Matter Networks Associated with Cognitive Dysfunction in Cerebral Small Vessel Disease"

_brainsci, 2023, doi:10.3390/brainsci13101359_

Round 1
Reviewer 1 Report
This paper reports on a singe study of topological organization of gray matter (GM) structural networks in patients with cerebral small vessel disease (CSVD) patients with or without cerebral microbleeds (CMBs) and compared to health controls. The authors found that compared with controls, CSVD patients all exhibited significantly increased global and local efficiency and decreased shortest path length. partial reorganization of hub distributions were seen between CSVD-c and CSVD-n patients. Regional topological properties in nonhub regions inside the default mode network and sensorimotor functional modules were significantly changed between CSVD-c and CSVD-n patients.
There are some minor issues the authors may wish to address:
1. Introduction – para 2 lines 5 and 12 – while ‘investigations’ and ‘studies’ were mentioned, only 1 reference each was provided
2. Experimental procedures – para - line 3 – please provide more information on the source and selection process for the ‘heathy subjects’. Para 2 – line 2 – spell out ‘MCI’ in full at first use
3. Results – diabetes mellitus among study subjects? Status of intracranial arteries eg severity and location and number of significant stenosis?
4. Table 3 – there are some spacing issues, maybe a smaller font would help. Please provide the ‘n’ at the top of the columns for CSVD-c, CVSD-n, HC; For ‘Gender’, to replace with ‘Sex, female (%)’ and provide the data accordingly.
5. Discussion – would the occurrence of the recent infarct result in temporary pathway disruptions or cognitive impairment that may soon resolve but affect the cognitive or other assessments?
Author Response
Dear Editors and Reviewers:
Thank you for your comments and providing good suggestions for my manuscript.
We have revised my manuscript according to your and the reviewer's suggestions. My explanation of the comments point-by-point is as follows.
Reviewer 1
This paper reports on a single study of topological organization of gray matter (GM) structural networks in patients with cerebral small vessel disease (CSVD) patients with or without cerebral microbleeds (CMBs) and compares them to health controls. The authors found that compared with controls, CSVD patients all exhibited significantly increased global and local efficiency and decreased shortest path length. Partial reorganization of hub distributions was seen between CSVD-c and CSVD-n patients. Regional topological properties in non-hub regions inside the default mode network and sensorimotor functional modules were significantly changed between CSVD-c and CSVD-n patients.
There are some minor issues the authors may wish to address:
- Introduction – para 2 lines 5 and 12 – while 'investigations' and 'studies' were mentioned, only 1 reference each was provided
Response: Thank you for your valuable advice. We have added the references (Introduction – para 2, Reference 11,13 – 15) in the revised manuscript.
- Experimental procedures – para - line 3 – please provide more information on the source and selection process for the 'healthy subjects'. Para 2 – line 2 – spell out 'MCI' in full at first use
Response: We have provided more information regarding the healthy older subjects in the manuscript (Page 6 – Experimental procedures – Subjects). We wrote the full name of MCI in the second paragraph of the Introduction, which is mild cognitive impairment (MCI).
- Results – diabetes mellitus among study subjects? Status of intracranial arteries e.g. severity and location and number of significant stenosis?
Response: Thank you for your valuable advice. Patients with diabetes were included in this study, and we ensured that these patients had good glycemic and blood pressure control and no acute complications.
- Table 3 – There are some spacing issues, maybe a smaller font would help. Please provide the 'n' at the top of the columns for CSVD-c, CVSD-n, and HC; For 'Gender', replace with 'Sex, female (%)' and provide the data accordingly.
Response: Thank you for making us aware of this. We have included the number (n) of study participants for each group in the tables, and we have replaced "gender" with "sex, female (%)" in the tables. Please see Table 1, page 11.
- Discussion – Would the occurrence of the recent infarct result in temporary pathway disruptions or cognitive impairment that may soon resolve but affect the cognitive or other assessments?
Response: Vascular risk factors increase the risk of stroke and cognitive decline. Studies show that cognitive decline can happen even before a stroke and accelerate after [1, 2]. Experts recommend using a validated global cognition screening test, with preference given to MoCA or Oxford Cognitive Screen based on available evidence [3]. In this study, we used the MoCA scale to assess the cognitive ability of patients, but patients with recent infarction were not included in this study.
References
- Levine, D.A., et al., Trajectory of Cognitive Decline After Incident Stroke. Jama, 2015. 314(1): p. 41-51.
- Zheng, F., et al., Progression of cognitive decline before and after incident stroke. Neurology, 2019. 93(1): p. e20-e28.
- Rost, N.S., et al., Post-Stroke Cognitive Impairment and Dementia. Circulation Research, 2022. 130(8): p. 1252-1271.

Reviewer 2 Report
This manuscript is well structured, the authors used complex and precise methods of clinical and statistical study presented in a comprehensible manner and their results were clearly exhibited in the last part of the article. However, the presentation of the final conclusions at the end of the article should be considered, for an easier overview.
Congratulations for your great work!
Author Response
Dear Editors and Reviewers:
Thank you for your comments and providing good suggestions for my manuscript.
We have revised my manuscript according to your and the reviewer's suggestions. My explanation of the comments point-by-point is as follows.
Reviewer 2
This manuscript is well structured, the authors used complex and precise methods of clinical and statistical study presented in a comprehensible manner, and their results were clearly exhibited in the last part of the article. However, the presentation of the final conclusions at the end of the article should be considered, for an easier overview.
Congratulations for your great work!
Response: We are honored to be highly valued by you. Thank you. We have now created a conclusions section with a summary explanation of the importance and relevance of the study.
Reviewer 3 Report
The authors presents the article entitled “Disrupted Gray Matter Networks Associated with Cognitive Dysfunction in Cerebral Small Vessel Disease”
This study investigates the disrupted topological organization of gray matter (GM) structural networks in cerebral small vessel disease (CSVD) patients with cerebral microbleeds (CMBs).
The article presents the following concerns:
-
Avoid using We, instead, use passive voice.
-
Reduce the percentage of plagiarism to less than 20%. According to Turnitin the manuscript has 45%.
-
It is necessary to mention the most outstanding quantitative results in the abstract.
-
It is recommended to number the sections and subsections.
-
At the end of the introduction, we will briefly mention the article's structure.
-
The controversy or problem to be solved concerning previous or similar works needs to be clarified.
-
Add a short introduction between sections and subsections.
-
I suggest to use the term “Equation” instead “Formula”.
-
I recommend to include the following two references to justify the sentence “networks based on structural MRI, the morphological GMV was calculated using VBM analysis, and then the network edges were defined as the statistical similarity of morphological distributions between different brain regions…”: Hyperconnected Openings Codified in a Max Tree Structure: An Application for Skull-Stripping in Brain MRI T1; Application of morphological connected openings and levelings on magnetic resonance images of the brain
-
The authors mention that 49 CSVD patients with CMBs and 121 CSVD patients without CMBs are more than twice as many volunteers in one class as in another. How was the imbalance in the database for the analysis of the results fixed or treated?
-
It is essential to mention in the section where the database describes the age range of the patients, as well as the other variables that could affect the results, such as percentages of men and women in the sample.
-
Put hyperlinks to figures, tables, and references.
-
It is necessary to improve the quality of the images.
-
It is necessary to describe and analyze the graphs and tables and not limit yourself to mentioning them in the text.
-
Table 4 and 6 needs to be edited better.
-
Add a comparative table with the results of this work and similar works.
-
The discussion is too long and contains parts of a conclusion. It is recommended to divide.
My biggest concern is about the level of plagiarism. For this reason, it is not possible to extend my recommendation for publication.
The following misspellings should be checked:
-
page 2: the infinitive “to produce” has been split by the modifier eventually. Avoiding split infinitives can help your writing sound more formal. Consider changing by “to produce clinical symptoms eventually…”
-
page 3: “had no prior understanding of…” Should be rewritten by “did not understand…”
-
page 14: The phrase “parts of the brain” may be wordy. Consider changing by “brain parts”.
-
Page 15: The use of “and/or” is severely frowned upon in formal writing. Consider using only one conjunction or rewriting the sentence.
Author Response
Dear Editors and Reviewers:
Thank you for your comments and providing good suggestions for my manuscript.
We have revised my manuscript according to your and the reviewer's suggestions. My explanation of the comments point-by-point is as follows.
Reviewer 3
Comments and Suggestions for Authors
The authors presents the article entitled "Disrupted Gray Matter Networks Associated with Cognitive Dysfunction in Cerebral Small Vessel Disease"
This study investigates the disrupted topological organization of gray matter (GM) structural networks in cerebral small vessel disease (CSVD) patients with cerebral microbleeds (CMBs).
The article presents the following concerns:
- Avoid using We, instead, use passive voice.
Response: We thank the reviewer for this advice. Throughout the article, we have rewritten sentences to a passive voice. Please refer to page 3, Abstract: Subjectwise structural networks were constructed from GM volumetric features of 49 CSVD patients with CMBs (CSVD-c), 121 CSVD patients without CMBs (CSVD-n), and 74 healthy controls.
- Reduce the percentage of plagiarism to less than 20%. According to Turnitin the manuscript has 45%.
Response: We're sorry to have given the impression of plagiarism. Given the comment raised by the reviewer for the plagiarism, the authors have taken this very seriously and corrected it line by line according to the plagiarism report provided. Some of the names of subjects and brief sentences in methods may show plagiarism to an extent, as these are very common sentences. The revised manuscript has been checked with turnitn plagiarism software, and the snapshot of the report is as under:
- It is necessary to mention the most outstanding quantitative results in the abstract.
Response: We thank the reviewer for this advice. We have added more details to the abstract. Please see page 3, line 6-17.
- It is recommended to number the sections and subsections.
Response: We thank the reviewer for this suggestion. All sections and subsections have been numbered.
- At the end of the Introduction, we will briefly mention the article's structure.
Response: Thank you for your valuable advice. We have briefly defined the structure of this article at the end of the Introduction. Please see the last paragraph of the Introduction, page 5.
- The controversy or problem to be solved concerning previous or similar works needs to be clarified.
Response: Thank you for your valuable and thoughtful comments. We feel sorry for the inconvenience brought to the reviewer, and we sincerely apologize for not explaining this part enough. Our previous study on the functional networks of CSVD patients found a significant decrease in global and local efficiency and a considerable increase in the shortest path length. However, our current study on structural networks discovered that the three topological properties mentioned earlier are opposite. Therefore, we are attempting to explain this as an adaptive optimization and reorganization of the brain structure network in CSVD patients, which allows for the proper functioning of the brain. We have rewritten this discussion section based on your helpful advice to ensure a clear description. Please see the seventh paragraph of the discussion.
- Add a short introduction between sections and subsections.
Response: Thank you for your suggestion. We have added corresponding instructions between sections and subsections.
- I suggest using the term "Equation" instead of "Formula".
Response: Thank you for your suggestion. We have replaced it now with "Equation" in the revised manuscript. Please refer to the supplementary materials.
- I recommend to include the following two references to justify the sentence "networks based on structural MRI, the morphological GMV was calculated using VBM analysis, and then the network edges were defined as the statistical similarity of morphological distributions between different brain regions…": Hyperconnected Openings Codified in a Max Tree Structure: An Application for Skull-Stripping in Brain MRI T1; Application of morphological connected openings and levelings on magnetic resonance images of the brain.
Response: We would like to thank the reviewer for this comment. In the revision, we have cited these two articles as you suggested (Introduction – para 3, Reference 22,23).
- The authors mention that 49 CSVD patients with CMBs and 121 CSVD patients without CMBs are more than twice as many volunteers in one class as in another. How was the imbalance in the database for the analysis of the results fixed or treated?
Response: We would like to thank the reviewer for this comment. An observational study involving nearly 4,000 participants aged 45 and above found that the prevalence of microbleeds increases with age, from 6.5% in the 45-50 age category to 35.7% in those aged 80 or older. The overall incidence rate for the cohort was 21.7%. There was no significant difference in microbleed prevalence between genders across all age groups. The average age of the study population was approximately 60.3 years, with females comprising 54.4% [4]. Our study population mainly consisted of individuals who were 40 years old or older. The average age of the CSVD-c patients was 63.69±8.37, the CSVD-n patients were 63.08±7.73, and the HC patients were 60.86±9.05. Regarding gender, the proportion of females was 38.8% in the CSVD-c, 48.8% in the CSVD-n, and 55.4% in the HC. However, there were no significant differences in age or gender composition among the three groups. The prevalence of cerebral microbleeds (CMBs) in this study was found to be 20.10%, which is in line with previous studies and the epidemiology of CMBs. Given the objective imbalance in sample size between groups, we also conducted the homogeneity test which is one of the prerequisite assumptions in the one-way ANOVA test, and we found the variance between groups were all homogeneous. Although imbalanced sample sizes may impact statistical efficacy, they do not significantly affect the accuracy of the model[5, 6]. We have added corresponding descriptions in section 2.6 and 3.1.
- It is essential to mention in the section where the database describes the age range of the patients, as well as the other variables that could affect the results, such as percentages of men and women in the sample.
Response: According to your suggestion, we have added the age range of the recruited volunteer in the Experimental Procedures section. Please see page 6, line 4.
- Put hyperlinks to figures, tables, and references.
Response: Thank you for your careful work. We have added the figures, tables, and references using hyperlinks so that the reviewer can comfortably see their quality.
- It is necessary to improve the quality of the images.
Response: We apologize that these critical parts have not been made clear. We have ensured that each image has a resolution of at least 300 dpi.
- It is necessary to describe and analyze the graphs and tables and not limit yourself to mentioning them in the text.
Response: We thank the reviewer for this professional advice. Related revisions were made in the Results section.
- Table 4 and 6 needs to be edited better.
Response: Thank you for your careful work. We have checked all the figures and tables and put them into the correct locations. Thank you for your correction.
- Add a comparative table with the results of this work and similar works.
Response: Thanks for the advice. The results and discussion section included a comparison table of relevant previous studies. Please refer to Supplementary Table S3.
- The discussion is too long and contains parts of a conclusion. It is recommended to divide
Response: Thank you very much. The discussion section has been condensed and divided into separate parts, with one part now as the conclusion.
My biggest concern is about the level of plagiarism. For this reason, it is not possible to extend my recommendation for publication.
Comments on the Quality of English Language
The following misspellings should be checked:
- page 2: the infinitive "to produce" has been split by the modifier eventually. Avoiding split infinitives can help your writing sound more formal. Consider changing by "to produce clinical symptoms eventually…"
Response: Thank you for your careful work. We have revised this sentence according to your suggestion. Please refer to the first paragraph of the Introduction section, page 4.
- page 3: "had no prior understanding of…" Should be rewritten by "did not understand…"
Response: Thank you for your careful work. We have revised this sentence according to your suggestion. Please refer to the second paragraph of the Subject section of Experimental Procedures, page 6.
- page 14: The phrase "parts of the brain" may be wordy. Consider changing by "brain parts".
Response: Thank you for your valuable advice. We have made corresponding changes to the revised manuscript based on your suggestions, pages 20-21.
- Page 15: The use of "and/or" is severely frowned upon in formal writing. Consider using only one conjunction or rewriting the sentence.
Response: Thank you for your valuable advice. We have made corresponding changes in the revised manuscript according to your suggestions.
References
- Levine, D.A., et al., Trajectory of Cognitive Decline After Incident Stroke. Jama, 2015. 314(1): p. 41-51.
- Zheng, F., et al., Progression of cognitive decline before and after incident stroke. Neurology, 2019. 93(1): p. e20-e28.
- Rost, N.S., et al., Post-Stroke Cognitive Impairment and Dementia. Circulation Research, 2022. 130(8): p. 1252-1271.
- Poels, M.l.M.F., et al., Prevalence and Risk Factors of Cerebral Microbleeds. Stroke, 2010. 41(10_suppl_1).
- Zhang, X., R. Li, and MD. Ritchie, Statistical Impact of Sample Size and Imbalance on Multivariate Analysis in silico and A Case Study in the UK Biobank. AMIA Annu Symp Proc, 2020. 2020: p. 1383-1391.
- Liang, Q., X. Yu, and S. An, [Influence of group sample size on statistical power of tests for quantitative data with an imbalanced design]. Nan Fang Yi Ke Da Xue Xue Bao, 2020. 40(5): p. 713-717.
Round 2
Reviewer 3 Report
The manuscript can be published